# Towards a Security Reference Architecture for NFV

**DOI:** 10.3390/s22103750

**Published:** 2022-05-14

**Authors:** Abdulrahman Khalid Alnaim, Ahmed Mahmoud Alwakeel, Eduardo B. Fernandez

**Affiliations:** 1Department of Management Information Systems, King Faisal University, Hofuf 31982, Saudi Arabia; 2Sensor Network and Cellular Systems Research Center, University of Tabuk, Tabuk 71491, Saudi Arabia; aalwakeel@ut.edu.sa; 3Department of Information Technology, University of Tabuk, Tabuk 71491, Saudi Arabia; 4Department of Electrical Engineering and Computer Science, Florida Atlantic University, Boca Raton, FL 33431, USA; fernande@fau.edu

**Keywords:** network function virtualization, virtual network function, cloud computing, reference architecture, security reference architecture, patterns, virtual machine environment, ETSI

## Abstract

Network function virtualization (NFV) is an emerging technology that is becoming increasingly important due to its many advantages. NFV transforms legacy hardware-based network infrastructure into software-based virtualized networks. This transformation increases the flexibility and scalability of networks, at the same time reducing the time for the creation of new networks. However, the attack surface of the network increases, which requires the definition of a clear map of where attacks may happen. ETSI standards precisely define many security aspects of this architecture, but these publications are very long and provide many details which are not of interest to software architects. We start by conducting threat analysis of some of the NFV use cases. The use cases serve as scenarios where the threats to the architecture can be enumerated. Representing threats as misuse cases that describe the modus operandi of attackers, we can find countermeasures to them in the form of security patterns, and we can build a security reference architecture (SRA). Until now, only imprecise models of NFV architectures existed; by making them more detailed and precise it is possible to handle not only security but also safety and reliability, although we do not explore those aspects. Because security is a global property that requires a holistic approach, we strongly believe that architectural models are fundamental to produce secure networks and allow us to build networks which are secure by design. The resulting SRA defines a roadmap to implement secure concrete architectures.

## 1. Introduction

To deliver network services to their customers, telecommunication service companies (Telcos) require a wide range of hardware appliances. With the growing demand from users for new network services, Telcos must devote more time and resources to deploying physical hardware and equipment for each network function, in addition to the need for highly skilled network technicians and operators to deal with the complexity of setting up and administering large networks. Furthermore, due to the rapid advancement of hardware, the network life-cycle is becoming shorter [1]. As a result, Telcos’ operational expense (OPEX) and capital expense (CAPEX) are expected to rise.

Network function virtualization (NFV) leverages virtualization technology to provision network functions (NFs) such as load balancers, switches, firewalls, domain name server (DNS), etc., which are built in software and offered as services; it allows NFs to be performed in virtual machines (VMs) using cloud infrastructure rather than physical infrastructure [2]. Transforming network appliances into virtual network appliances reduces the need to install, maintain, and acquire special hardware at the customer premises and reduces energy consumption and cost. Also, the agility of NFV encourages network operators to adopt it due to the capability of self-management of network services, and the ability to create a new market that facilitates the development of new businesses. Moreover, NFV promises the following benefits [3]:Self-sufficiency: the software is no longer linked to the hardware. As a result, they will evolve independently of one another.Flexibility and speed: by decoupling software from hardware, it is possible to reassign and share infrastructure resources, allowing different functions to be performed at different times. As a result, network operations and their connections may be deployed more quickly and with greater flexibility.Scalability: in regular legacy network systems, Telcos have to be up to date with new network standards and requirements, which requires time, planning, and money. However, in NFV, decoupling software from hardware allows for dynamically scaling the actual performance of virtualized network functions with finer granularity and minimal effort.Reduced energy consumption: with the ability to scale up or down resources, Telcos will be able to reduce the OPEX needed to run network devices. Similarly, energy consumption at the customer end will be reduced significantly due to not having to install dedicated hardware to deliver network functions.Speed to set-up the network: the deployment and configuration of network services is much faster in NFV.

In October 2012, network function virtualization was introduced as a new concept by the European Telecommunications Standards Institute (ETSI). The ETSI, with the contribution of telecommunication vendors, presented the first architecture for NFV [3]. Their architecture consists of three main architectural components which are: network function virtualization infrastructure (NFVI), which comprises all the hardware and software components to support the execution of the virtualized network functions; virtual network functions (VNFs), which are software implementations of network functions; and management and network orchestration (MANO), which covers the VNF lifecycle management and orchestration of physical and software resources. 

Despite its advantages, NFV increases system complexity and is susceptible to new threats [4], which makes it important to understand its security issues. Since they rely on software, the network functions in NFV can be configured and controlled by external entities, such as a third-party provider, or a consumer. This makes manipulating the network service easier than in the traditional network infrastructure. In general, the attack surface of NFV is considerably increased compared with the traditional network infrastructure. In addition to the network resources (switches, routers, load balancers, etc.) in the traditional network infrastructure, the whole virtualization environment including live migration and multi-tenant infrastructure could be exposed to more threats than in the traditional networks. The fact that NFV is a fundamental technology of 5G networks indicates that the importance of NFV security will increase significantly. Another important technology, often used to implement NFV, is SDN. Security in 5G and SDN has been studied in several works [5,6,7]; NFV requires the support of cloud computing, and its threats have been studied in many publications. We concentrate here only on modeling some of the security aspects of NFV.

Because security is a global property that requires a holistic approach, we strongly believe that architectural models are fundamental to produce secure networks and allow us to build networks which are secure by design. We therefore start by finding the possible threats through analysis of NFV use cases. The use cases serve as scenarios where the threats to the architecture can be enumerated. Representing threats as misuse cases that describe the modus operandi of attackers, we can find countermeasures to them in the form of security patterns, and we can build a security reference architecture (SRA). Security patterns are encapsulated solutions to security problems, while misuse patterns describe attacks from the point of view of the attacker. Until now, only imprecise models of NFV architectures existed; by making them more detailed and precise it is possible to handle not only security but also safety and reliability, although we do not explore those aspects. The resulting SRA defines a roadmap to implement secure concrete architectures. To our knowledge, there is no work in which architectural modeling, i.e., patterns and reference architectures, has been applied to describe and analyze NFV systems except [8], which developed an architectural pattern that describes the general concept of NFV, its advantages and disadvantages. In summary, our security reference architecture is the only detailed SRA for NFV. Given the large scope and repetitive features of an SRA, we only show a partial SRA and indicate how to complete it.

Further, using our survey of the main security threats of NFV [9], we analyze use cases to enumerate their threats. Then, we show how some of the identified threats may lead to several misuses of information and describe three of them using misuse patterns [10]: privilege escalation [11], VM escape [12], and distributed denial-of-service [13]. These misuse cases demonstrate how we select security patterns as countermeasures, which leads to the SRA. These patterns have been published by us in conferences; here, we describe summaries of them to show how they contribute to the functions of the SRA. The use of patterns is well accepted in secure software development and there are catalogs that make their use convenient [10]. Other quality factors such as safety and reliability are outside our scope, their methods and solutions are different from security. 

The rest of the paper is structured as follows: Section 2 provides background about the NFV architecture proposed by ETSI, and on patterns and reference architectures. Section 3 shows the steps toward building a security reference architecture for NFV by first enumerating the threats through misuse activities, and how these threats lead to several misuses; three misuse patterns are shown. The section also shows security mechanisms that mitigate the identified misuses, and as a result we produce a partial security reference architecture for NFV. Section 4 shows some validation aspects of our models. Section 5 considers related work. Section 6 provides conclusions and future work. 

## 2. Background

### 2.1. NFV Architecture

The network function virtualization infrastructure (NFVI) is the NFV’s foundation platform, containing both hardware and virtual instantiations that make up the infrastructure on which VNFs are deployed, managed, and executed. The NFVI can be considered as a component of cloud infrastructure-as-a-service (IaaS), which allows cloud providers to establish virtual data centers (VDCs) [14], which comprise all of the virtualized computing, storage, and networking resources required to operate as a physical data center. These VDCs are given to NFV providers, who use them to provide network services to customers. The resources of a VDC offered to a specific NFV provider should be segregated from those of other providers; this isolation allows NFV providers to share the same cloud infrastructure in a secure manner. When it comes to NFV services, the VNFs are deployed over virtual machines (VMs) within a VDC. 

The NFVI is made up of three basic components, as depicted in Figure 1. First, there are the hardware resources, which include compute facilities, which are typically commercial-off-the-shelf (COTS) appliances, storage hardware, which could be in the form of direct-attached hard disks, external storage area networks (SAN), or network-attached storage (NAS) [15], and network hardware, which could be switches/routers that provide processing and connectivity capabilities to VNFs via the virtualization layer. The virtual machine monitor (VMM) (also known as the hypervisor) is part of the virtualization layer, which sits on top of the hardware resources layer and performs three main functions: decoupling virtual resources from underlying physical resources, providing isolation among VMs, and emulating hardware resources [16]. Third, the virtual infrastructure sits on top of the virtualization layer and incorporates virtualized resources, such as virtual machines, virtual storage, and virtual networking, which are abstractions of hardware resources.

VNFs (virtual network functions): VNFs are software packages that represent the implementation of legacy network functions on the NFVI. Packet data network gateways (PGW), residential gateways, firewalls, and other internal components (VNFCs) could make up a single VNF [3,17]. A VNF, on the other hand, could only have one component, despite the fact that a single VNF might be deployed and dispersed across multiple VMs [3]. Further, a network graph is a collection of VNF services that together give the intended service to the customer, i.e., they might create a virtual network based on the available VNFs. Telcos’ virtual network services are typically made up of many VNFs that are tailored to the demands of their customers.

NFV management and orchestration (MANO) is in charge of managing and orchestrating all virtualization-specific operations necessary throughout the VNF’s lifespan, from merging several services into a single VNF package to mapping this service to consumers upon request. The MANO also manages any VNF failures that may occur, as well as maintaining state information for each VNF in the service. Furthermore, the MANO is in charge of establishing communications amongst the various VNFs that make up the network graph. The MANO is made up of three main functional blocks:Virtual infrastructure manager (VIM): is responsible for managing and controlling the interaction of the VNFs with the NFVI resources. The VIM performs resource management functions, such as keeping an inventory of software as well as management and orchestration of resources. The VIM is also responsible for collecting and logging information to check for faults, as well as collecting information for the purpose of capacity planning, performance monitoring, and performance optimization [3].VNF manager (VNFM): is responsible for managing and monitoring the VNF through the element management system (EMS), which includes scaling, changing operations, and adding new resources to the VNF, as well as communicating the states of VNFs to the other functional blocks that create the NFV architecture.Orchestrator: provides the necessary resources and networks needed to set up cloud-based services and applications, including the use of different virtualization software as well as hardware [15].

### 2.2. Patterns and Reference Architectures

A pattern is a solution to a recurrent problem in a specific context [18]. Patterns differ based on their purpose and the issue they solve. Security patterns are used to build secure systems by describing ways to control threats, patch vulnerabilities, and provide security attributes [10]. Design and architecture patterns are used to build flexible and extendible systems. Misuse patterns are used to describe how attacks are carried out from the attacker’s perspective [19]. They also define the environment in which the attack is carried out, what security mechanisms are required as countermeasures to stop it, and where forensic information can be found to trace the attack once it has occurred.

A reference architecture (RA) is a generic abstract software architecture for analyzing, designing, and understanding complex systems. An RA includes a set of stakeholders, use cases, and a diagram that outlines system components, their functionalities, and their interdependencies but does not include implementation details [20]. Security mechanisms can be incorporated in appropriate places within the RA to manage identified threats, thus defining a security reference architecture.

Patterns are abstractions of best practices; they do not propose new solutions. Because of this, pattern papers do not include implementations or experiments, the originality of patterns and reference architectures is in the completeness and fidelity of the models with relation to existing systems; the idea is to reuse this knowledge and experience for new designs and to evaluate existing designs. SRAs are similar in intent and use and are similarly validated. 

## 3. A Security Reference Architecture for NFV

In [9] we surveyed some threats and their possible misuses. Threats and their effect can be described using misuse patterns. Misuse patterns describe how attacks are performed from the point of view of an attacker [19]; they define the environment where the attack is performed, what security mechanisms are needed as countermeasures to stop it, and how to find forensic information to trace the attack once it happens. We show here three misuse patterns for three threats in NFV: privilege escalation [11], VM escape [12], and denial-of-service [13]. These misuse patterns are considered the first step toward building a security reference architecture (SRA) because they show what defenses (security patterns) can prevent the attack. As shown in the pattern diagram in Figure 2, a misuse pattern defines a threat to the NFV RA, and it can be mitigated by a security pattern. We can obtain an SRA by adding security patterns to the RA to control all the identified threats. In other words, an SRA is an RA where all its identified threats have been controlled by adding security patterns. If there are many threats, it is useful to perform a risk analysis where the probability of occurrence and the impact of the threat are evaluated; however, the standard methods for risk analysis can be used and we do not discuss this stage here. For one thing, the actual selection of threats depends on the specific application.

### 3.1. Finding Threats through Misuse Activities

We present now a use case of NFV to show how we find threats in the system. Use case activity diagrams show the activities involved in executing the use cases. Each activity can create or use assets that can be attacked; dashed rounded rectangles represent threats and dashed lines represent misuse-related control flows. We identify threats for NFV through a systematic analysis of use case activities, considering each activity in every use case. After enumerating threats, we select appropriate security policies to mitigate these threats; we also connect the threats to their source to indicate if they are from an insider or an outsider attacker [21]. Since all the use cases represent all the interactions with the system and we examine all their activities, we can find almost all the threats, which means that if we can neutralize them, the system is secure in a practical sense. Threat Analysis of Use Cases “Request Modify a VNF” and “Consume a Network”.

One of the main features of NFV is its flexibility compared to legacy networks. In NFV, consumer-operators (Con-Ops) can request at any time a modification of the resources they are using according to their needs, whether it is scaling the network up or down, which makes this use case one of the fundamental use cases of the NFV life cycle. The process of modifying resources in NFV is done merely by sending a request from the Con-Op to the NFVP operator, which in turn alters the network to match the consumer’s needs and sends a modification acknowledgment back to the Con-Op. However, during this process, several threats are possible. For example, the Con-Op, who could be an impostor, compromises the VNF the consumer is using to send a malicious hypercall that leads to taking control of the hypervisor, which results in several misuses, such as escalating the privileges of the attacker’s VM, or letting the VM escape from the hypervisor control. Figure 3 illustrates the activity diagram for the use cases request to modify a VNF (UC13) and consume a network (UC20) showing some possible misuses.

Through a systematic analysis of the flow of events in the activity diagram of Figure 3, we have identified several threats that could affect the network service. Table 1 shows an analysis of each action in the previous activity diagram based on two aspects: the main security attributes that are confidentiality (CO), integrity (IN), availability (AV), and accountability (AC), and the source of threat whether it is from an authorized insider (AIn), unauthorized insider (UIn), or from an outsider (Out).

After analyzing the use case activities and discovering different misuses, we can recommend some appropriate mitigation mechanisms that can be implemented and included in the system in order to mitigate these misuses. Table 2 shows the list of the identified misuses and their mitigation mechanisms. The fact that the authenticator is designed to stop unauthorized access to the system is proof that this action can be stopped.

### 3.2. A Misuse Pattern for Privilege-Escalation-Based Misuses

Intent

VMs are created and managed by the hypervisor, which has rights to fulfill their hypercall requests and ensures isolation among them. An attacker, who can be a consumer, may escalate the privilege of his VM allowing him to perform hypervisor privileged operations that may lead to several misuses, such as unauthorized access to hardware, create malicious VMs, and terminate co-resident VMs.

Problem

In order to perform the misuses, the attacker runs a malicious application in his VNF that sends a malicious hypercall to the hypervisor. The misuse can be done by exploiting the following vulnerabilities:VMs can send any type of hypercalls, whether they are legitimate or malicious, to the hypervisor.Hypercalls are low-level kernel requests for processing and resource access, and distinguishing genuine from malicious hypercalls can be challenging.Because the network service is hosted in a sharable environment, if one VNF is compromised, the other co-resident VNFs may be affected.The emergence of new attacks like return-oriented programming (ROP) attack, which allow attackers to change data in the hypervisor that modifies VM privilege levels.

Solution

An attacker would be able to request network services if he had a valid account. Then, an application running on top of his VNF can send process and resource requests to the hypervisor via hypercalls, thus allowing the attacker to take advantage of that application to run a malicious code. These hypercalls are processed using the hypervisor files. Exploiting the CVE-2011-1583 vulnerability, which grants attackers control of the hypervisor and modifies their VM privilege level [22], is one way that attackers can elevate their VM privilege level. Another method is to use a malicious hypercall, such as the one developed in [23] and applied to the Xen hypervisor, as we will demonstrate later.

Each VM has a domain structure stored in the hypervisor; this structure contains basic information about the VM, such as domain_id, which is an identification number for the VM, is_privilege, which indicates the privilege level of each VM, and next_in_list, which is used to link these domain structures together in ascending order by their domain_id. The Xen hypervisor, for example, creates a parent VM (domain 0) that creates child VMs (domain 1, domain 2, etc.) by traversing dom0, the domain structure of dom1 can be identified by knowing the domain_id of dom1. 

The misuse begins when a malicious hypercall is delivered to the hypervisor with an overflow in its stack buffer, allowing the malicious application to access the hypervisor address space and subsequently launching a return-oriented programming (ROP) attack. A ROP is an attack that alters existing codes in the hypervisor memory space rather than writing new ones. The result of the ROP attack is escalating the privilege of attacker’s VM by changing the is_privilege value from 0 to 1. A successful attack allows attackers to perform hypervisor operations such as terminating virtual machines, resulting in a denial of service, and gaining unauthorized access to hardware resources, thereby compromising the entire virtualization environment. This attack scenario is applied in a cloud computing environment, and the NFV is able to stop VNFs co-existing with a malicious VNF or even create more malicious VNFs [24].

Structure

Figure 4 shows a class diagram for privilege-escalation-based misuses in NFV [11]. Through the portal, the consumer connects to the NFV service. He can also use the APIs to access VNF functions. The VIM is a MANO software unit that is in charge of managing the NFV infrastructure, including the hypervisor. The hypervisor manages the hardware resources required to provide network services, as well as mediating access to them. It also controls the VMI repository, which stores VM images (VMIs), creates VMs, assigns them to consumer accounts, and manages the hardware resources required to provide network services. A VMI is used to create VMs. The hypervisor allocates hardware resources to a VM after it is launched. Many VMs can have a set of VNFs assigned to each consumer. Hypercalls are system calls sent by virtual machines to the hypervisor, which executes them in hardware.

Dynamics

Misuse case (MC) 1: Unauthorized access to hardware based on privilege escalation of an attacker’s VM (Figure 5).

Summary: the attacker, who is a consumer, runs a malicious application in his VNF to send a malicious hypercall to the hypervisor. The application accesses the hypervisor address space and launches an ROP attack that modifies the is_privilege value of the attacker’s VM from 0 to 1 (unprivileged to privileged). The success of this escalation allows the attacker to illegally access hardware resources and retrieve confidential data.

Actor: attacker

Precondition: the attacker has a valid account and active network services.

Description:The attacker first runs a malicious application in his VNF.Using the malicious application, a malicious resource request is sent as a hypercall.The malicious hypercall is forwarded to the hypervisor through the VM.The hypervisor receives the malicious hypercall request and fulfills it. In this case, the malicious application accesses the hypervisor address space and launches an ROP attack. The result of the ROP is escalating the privilege of the attacker’s VM by changing the is_privilege value from 0 to 1.The hypervisor escalates the VM privilege level.The attacker is notified that the VM has successfully been escalated.The attacker is now able to illegally access hardware resources.

Postcondition: the attacker controls the hypervisor, and has direct access to the hardware resources, with which he can perform malicious operations.

Consequences

A successful attack leads to the following consequences:The attacker is able to compromise the system and its data because he is able to illegally access hardware resources.The attacker can completely disrupt the network services (DoS), preventing NFV consumers from using the service.Escalating the privilege of the attacker’s VM enables him to perform hypervisor operations such as directly accessing hardware resources and jeopardizing the system servers, creating, starting, stopping, migrating, and terminating victims’ VMs.The attacker may be a competitor in the network service market and aims to damage the reputation of the NFV provider as their service has been disrupted and will appear to have security issues.

Countermeasures

VM privilege escalation can be mitigated using the following countermeasures:System bugs are patched by hypervisor vendors; therefore, the attack can be mitigated using vendor patches in [25] if the attacker exploits this vulnerability that leads to privilege escalation [22].Using some security tools that mitigate ROP attacks such as G-Free [26], HyperCrop [27], HyperVerify [28], ROPecker [29], as well as the hardware virtualization mechanism proposed in [30].Using security schemes including hypercalls control and authenticated hypercalls that help to verify the authenticity of hypercalls [31].

### 3.3. A Misuse Pattern for Compromising VMs via Virtual Machine Escape

Intent

The hypervisor creates and manages VMs and ensures isolation among them. An attacker, who can be a consumer, aims to compromise this isolation by executing arbitrary codes that let a VM escape from the hypervisor control and undertake possible misuses.

Problem

How can an attacker violate the isolation that the hypervisor provides among its VMs and the hardware, thereby accomplishing a VM escape from the hypervisor control? The attack can be done by exploiting the following vulnerabilities: The hypervisor is a software that includes various components, such as: drivers, schedulers, and hypercalls table. The security of hypervisor is difficult to maintain due to its design complexity [32].The hypervisor is one of the most critical components in the virtualization environment since it mediates the virtualized instances and hardware. As a result, a faulty hypervisor configuration can lead to a number of vulnerabilities that attackers can use to launch attacks, such as VM escape [33].VMs are interconnected with the host OS through the hypervisor; if there is a lack of isolation, attackers will be able to break into the host OS.The existence of many vulnerabilities that lead to VM escape proves that the virtualized environment is not robust enough even if these vulnerabilities have already been patched.Hypervisors send and receive hypercalls from VMs. These hypercalls are low-level requests for basic processing and resource access and it is difficult to differentiate between legitimate and malicious hypercalls; therefore, attackers can take advantage to send malicious requests to the hypervisor.The network service is hosted on a shared environment; if a VNF is compromised, that may affect the other co-resident VNFs.

Solution

An attacker is able to request network services if he has a valid account. In a running up service, he could run malicious procedures in an application executing on top of his VNF that sends crafted network packets to exploit a heap overflow with a compromised virtualization process controlled by the attacker. Usually, crafted packets are used to perform tests on networks; however, attackers may create packets to carry out attacks and to exploit vulnerabilities in a network. In a practical scenario [34], these packets are sent through hypercalls. As a result, the attacker will be able to execute arbitrary code on the hypervisor to gain access to the host OS, resulting in letting a VM escape from the hypervisor control. In addition, the escaped VM could interface directly with the hypervisor, bypassing any isolation, and will have access to the other victims’ VMs that are co-resident with his VM. This allows the attacker to gain access to each victim’s virtual machines and carry out a variety of malicious actions, such as disseminating confidential information, stopping the hypervisor and the VMs operating on top of it, and so on.

Structure

As shown in Figure 6 [12], to request NFV services, the consumer uses the portal to connect to the NFV service. He can also use the APIs to access network functions. The VIM is a MANO software unit that is in charge of managing the NFV infrastructure, including the hypervisor. The hypervisor controls and mediates access to the VMI repository, which stores VM images (VMIs), builds VMs, assigns them to users’ accounts, and manages the hardware resources required to offer network services. A VMI is used to construct each VM. The hypervisor allocates hardware resources to a VM after it is launched. VMs can contain a large number of VNFs that are assigned to each consumer, or a VNF can use several VMs. Hypercalls are system calls sent by VMs to the hypervisor, which performs them in the host. In the attack scenario outlined above, the NFV consumer (attacker) runs malicious operations on a VNF application, allowing him to send designed network packets as hypercalls to the VM hosting his VNF. The VM will be compromised, and many of the co-residents’ VMs will be affected as well. The attacker can also execute arbitrary code on the hypervisor, allowing him to gain access to the host OS files.

Dynamics

MC2: compromise a victim’s VMs via VM escape (Figure 7).

Summary: a VM escape attack is performed by the attacker, who is a consumer.

Actor: attacker

Precondition: the attacker has a valid account and active network services.

Description:The attacker first runs a malicious application in his VNF and compromises it by gaining access to the VM’s operating system.Using the compromised VNF, the attacker sends a crafted network packet to the VM in order to exploit a heap overflow.Arbitrary codes are executed on the hypervisor resulting in a VM escape. These codes enable the attacker to gain access to the host OS.The attacker is notified that the VM has successfully escaped from hypervisor control.As a result, the attacker can now control the VM.The attacker can read/write the hypervisor data through the controlled VM.Now, the attacker is able to compromise the victim’s VM using the controlled hypervisor.

Postcondition: the attacker compromises co-resident victims’ VMs that may lead to several misuses of information. 

### 3.4. A Misuse Pattern for Distributed Denial-of-Service Attack in NFV

Intent

An attacker intends to exhaust network resources and impact service availability to legitimate users by sending a huge number of requests from a domain name server (DNS). 

Problem

How can an attacker flood a target (a network service of a consumer) with a large number of DNS requests consuming most of its resources, e.g., bandwidth, thus achieving a DDoS attack? 

The attack can be performed by exploiting the following vulnerabilities:NFV is a recent technology, and its security infrastructure has not been tested enough in the wild, which raises the possibility that vulnerabilities may lead to several threats including denial of service.The urgency to adopt NFV services may have let NFVPs focus more on profits without hardening the security of their infrastructure including the DNS server configurations [35].The network service is hosted on a shared environment running on top of VMs; if a VNF is compromised due to misconfiguration, malware infection, or by exploiting a vulnerability in an old version of software running on it, a huge amount of traffic can be generated from the compromised VNF and sent to other co-resident VNFs running on the same hypervisor or even on different hypervisors, or to DNSs [36].The NFV environment provides network functions with a higher degree of flexibility and configuration possibilities than traditional architectures; therefore, there are more ways to misconfigure the network functions, which increases the attack surface and opens new avenues to compromise the system [37].The elasticity of the NFV environment enables network resources to rapidly scale up or down. In case of the DNS amplification DDoS attack, attackers can take advantage of this property to amplify the attack when multiple vDNSs will be created due to the traffic load and will produce a huge number of DNS replies to the victim. This scenario is possible in NFV environments, and we demonstrate it in later sections.Domain name systems, especially public DNSs, are designed to respond to any request where an attacker can turn a few DNS requests into much larger payloads. Attackers can leverage this amplification effect to launch a DDoS attack.

Solution

The attack is possible when the attacker floods the network resources with a large number of DNS requests using spoofed IP addresses. First, we assume the attacker has a botnet of infected devices [38], and a list of victims’ IP addresses. The attacker spoofs the IP addresses of victims’ machines and uses the botnet, which is controlled by the command-and-control server, to launch a large number of malicious DNS requests (UDP packets). These requests are sent to a VNF (a vDNS). The orchestrator realizes that the traffic load on the vDNS is above the normal threshold, and in turn the hypervisor initiates new VMs to scale-out additional vDNSs to accommodate more requests. This elastic and scalable nature of NFV will make multiple vDNS recursively respond to the victims, and in effect will receive amplified DNS responses, which can ultimately result in service disruption for the victims’ web servers. This practical scenario is described by [34] and it is possible in any system, including NFV.

Structure

Figure 8 shows a class diagram for distributed denial-of-service attack in NFV, showing the units compromised by this attack. The command-and-control server (CC) is a centralized management platform controlled by the attacker used for orchestrating the attack. Botnet is a network of attacking machines, which the attacker uses to launch a high number of malicious DNS requests that contain spoofed IP addresses matching the victims’ IP addresses. The consumer accesses the VNF functions using their APIs. The vDNS is one of the VNF functions. The hypervisor controls the VMI repository that stores VM images (VMIs), creates VMs, and manages the NFVI resources necessary to provide network services. Each VM is created using a VMI, and VMs can contain several VNFs. Once a VM is launched, the hypervisor assigns resources to it. The MANO is a management and orchestration (MANO) unit that has several roles, such as reporting status information to the hypervisor through the VIM, as well as monitoring the VNF through the VNFM.

Dynamics

MC3: distributed denial-of-service attack in NFV using DNS amplification attack (Figure 9).

Summary: an attacker uses spoofed IP addresses to send a large number of DNS requests to vDNSs, that in turn send amplified responses to victims, resulting in service unavailability or disruption.

Actor: attacker

Precondition: the attacker controls a botnet of infected devices and a list of victims’ IP addresses.

Description:The attacker first sets up the command-and-control server (CC) and activates it.Through the command-and-control server, the attacker activates the botnet and sends attack commands to the botnet to launch a high number of DNS requests to a vDNS.Meanwhile, the MANO is monitoring the VNF and realizes that the traffic load is higher than the normal threshold and reports it to the hypervisor.As a response to the high traffic load on the vDNS, the hypervisor initiates additional VMs to scale-up additional vDNSs to accommodate more requests.Accordingly, amplified DNS requests are recursively sent from the vDNSs to the victims’ web servers, which results in service unavailability or disruption.

Postcondition: the victims’ VMs will be disrupted or become unavailable. 

### 3.5. Securing the NFV Service

VNFs are considered the core of the NFV system; therefore, they are a high value target for attackers who intend to compromise the network. We now consider some security mechanisms to protect the network service from the identified misuses from Table 2. Figure 10 shows a partial security reference architecture (SRA) after adding some security mechanisms, shown in blue, in order to protect it from MC1 and MC2. The SRA is based on two use cases: UC13 (modify a VNF) and UC20 (consume a VNF). The authenticator [5] prevents outsider attackers from accessing the system, which can stop the threat T1. A security logger [5] is used in T2 and T6 to record and keep track of any activity within the network resources, and a security auditor for forensics. The filter scans the VNFs in order to remove malicious codes such as wrong size parameters, wrong type parameters, or wrong operations, which can stop the threat T3. The secure channel [10] mitigates T4, which provides integrity and confidentiality for the network service. Role-based access control (RBAC) [10] is used to control access to virtual resources based on predefined rights, and can be used to mitigate T6. The controlled virtual address space (CVAS) pattern [10] is used to limit and control access by processes to specific areas of their virtual address space; this can be used to mitigate T5, and prevent privilege escalation. 

#### 3.5.1. Security Evaluation of the Security Reference Architecture of NFV

In previous sections, we analyzed the use case activities of UC13 and UC 20 and identified the possible misuses in each activity. Then, we applied mitigation mechanisms that could defend against these misuses. Now, we need to evaluate our partial SRA to check whether it covers the identified misuses. We will use the misuse patterns described earlier to test our model. In Figure 5, we showed how an attacker (Con-Op) can insert and run malicious codes on the VNF in order to send a malicious request to the hypervisor that results in modifying the privileges of the VM that hosts the attacker VNF, or to let the VM escape from hypervisor control, as shown in Figure 7. Therefore, we use the sequence diagram in Figure 11 to show how the security mechanisms can stop the threats identified in MC1 and MC2. In Figure 11, it can be seen that the attacker (Con-Op) will be authenticated during login to make sure only legitimate users access the NFV system. Even if the attacker has a valid account, his attempts to insert malicious codes on the VNF will be handled by the filter that scans and removes malicious codes. Similarly, we can show that the other threats can be handled by corresponding security patterns.

As shown above, from misuse patterns we can find security patterns that will stop misuses from the attackers. Security patterns are defined with the purpose to control specific attacks. All the patterns we use here have been validated by their presentation in pattern conferences and in practice and are catalogued in [10]. If a threat is identified in a reference architecture and that threat can be handled by a pattern, we consider that if the pattern is instantiated in the architecture it will be able to stop the corresponding attack. The analysis of all the use cases would give us confidence that all the important threats have been identified. Figure 11 shows how the insertion of security patterns can stop the misuses which are possible from two misuse cases. 

## 4. Validation of SRAs

Architectural modeling has been proven to be a powerful mechanism to present and understand complex systems. However, SRA is an abstract model, and such models cannot be evaluated in terms of performance, reliability, or security using traditional experimentation or testing methods [39]. Building a prototype is a complex endeavor; furthermore, if not secure, it cannot invalidate the SRA because the prototype may have implementation flaws. On the other hand, other criteria can be used to validate the SRA and security patterns. Security evaluation, showed in Section 3.5.1, is an aspect of this validation since we can show by argument that our model is secure. We can achieve further validation of abstract models by comparing models to models of other commercial NFV systems; we explicitly compare here the fundamental components of our NFV SRA with the elements of these systems; keeping in mind that some of these systems use additional components to support their functions that are not essential to NFV basic service. This provides a completeness validation.

Our SRA was developed deductively from an analysis of threats. No industrial company has published an SRA for their NFV products. Most of the NFV project reports focus on the deployment of NFV, without describing ways of securing their architectures. This could be because the NFV paradigm is still a recent concept in the networking industry, and it will take time to define its security requirements. As a result, our SRA provides a guideline to industry to show what security defenses they should include in their architectures. We can still compare our SRA to some frameworks and the completeness and accuracy of our SRA is validated by comparing its components and security measures to them. 

We compared our models with the ETSI security framework [40,41], Cloud Security Alliance (CSA) security framework [4], and Red Hat NFV platform, which is based on OpenStack cloud computing platform [42,43,44]. We chose the ETSI security framework because we used their architectural framework to build our models. The CSA security framework provides comprehensive security measures; it is appropriate to include it in this validation. Further, the Red Hat NFV platform is one of the most widely used platforms found in the literature.

Table 3 shows the comparisons between our SRA and the selected frameworks. We list all of the main components of our SRA and its security mechanisms as columns, and the selected NFV security frameworks are presented in rows. Each component of our SRA is matched with the security frameworks, and a mark of “✓” is placed if the component is found in that particular security framework.

Our SRA contains several components derived from the reference architecture of NFV [45], we list only some of them, which we consider the main components that are necessary to ensure the operability of the network service, and they should be found in every NFV framework. As we mentioned earlier, our SRA is in ongoing stage and does not cover all of the needed security for the NFV system; however, it does mitigate the threats described in Section 3. 

As shown in Figure 10, the SRA architectural components are VNF, VM, hypervisor, API, and VIM; we excluded the portal, account, and customer components. As can be seen in Table 3, the ETSI, CSA, and Red Hat frameworks contain these components in their NFV security frameworks. It has been inferred that these components are necessary in every NFV platform to ensure the operability of NFV service. The SRA security components are authenticator, filter, security logger/auditor, RBAC, CVAS, and secure channel. The authenticator is used to control access to the NFV system, the security logger/auditor is used to keep track all of security activities, and RBAC is used to enforce authorization rights based on roles; all of the three frameworks use these three security mechanisms in their systems. The filter is used to scan the VNFs in order to remove malicious codes, and it is used by all the three frameworks. The filter also can be found in ETSI [17] and Red Hat in a form of a web application firewall (WAF), which filters and monitors the traffic coming from web applications. The secure channel provides integrity and confidentiality for the network service; the three frameworks use it, but they may differ in the actual mechanism; for example, Red Hat uses transport layer security (TLS) to protect communication between users and their platforms [43]. The CVAS controls the use of rights of the SRA and is able to mitigate privilege escalation and DDoS attacks. Some of the three frameworks use additional security mechanisms to mitigate such attacks; for example, CSA uses three security functions to filter malicious DDoS traffic before it reaches the targeted network service [4].

It can be found that our SRA model is fairly accurate as its components and security mechanisms match well those of the NFV security frameworks. Indeed, these NFV frameworks contain additional security measures, but as we mentioned earlier, our SRA is partial and does not cover the security of the whole NFV system. The validation can be further extended when new NFV security frameworks are published. 

## 5. Related Work

Several surveys of threats and other security aspects of NFV have appeared, starting from [34], and including our own [9]. More recently, Pattaranantakul et al. [46] analyze security threats of five NFV use cases to build a taxonomy of threats in the network layers, then compare security mechanisms, to end with recommendations to secure NFV services. Farris et al. [47] considered threats when NFV is used in IoT systems, providing an analysis of the security effect of introducing NFV, describing strategies to monitor, protect, and react to IoT security threats. Their paper also compares defense approaches in IoT environments to conventional security countermeasures. Wu et al. [48] analyze the state of security for NFV and propose security practices for an NFV-based management and control ecosystem; they also identify ongoing research challenges and open security issues for NFV. 

The most recent survey considers NFV use in 5G [7,49]; it explores the complexity of the 5G ecosystem for which it derives a 5G telecommunication business model, which is used to identify new aspects of its attack surface and enumerates security threats introduced by the multiple business collaborations supported by NFV. That paper proposes a three-dimensional threat taxonomy for NFV-based 5G networks relating NFV architecture to deployment use cases for different 5G business collaboration scenarios. The proliferation of IoT devices brings a variety of attacks because they are an integral part of a network slice and require specific QoS and security requirements. Differently from earlier surveys, they investigate threats at inter-layer, intra-layer, and inter-administrative domains, and also review existing surveys on cloud computing, NFV, and SDN security as well as some NFV security projects. Ideas to enhance security in these networks are also provided.

Other works add security to NFV networks. Basile et al. [49] introduce an approach towards automatic enforcement of security policies in NFV networks and for dynamic adaptation to network changes. The proposed approach transforms high-level security requirements into configuration settings and optimal selection of network security functions (NSFs). These models are built on a formalization of the NSF capabilities, which describe what NSFs can do for security policy enforcement purposes. A network security function (NSF) is a function used to ensure integrity, confidentiality, or availability of network communications, to detect unwanted network activity, or to block or at least mitigate the effects of unwanted activity. They consider this work as the first step towards a security policy aware NFV management, orchestration, and resource allocation system—a paradigm shift for the management of virtualized networks—and it requires minor changes to the current NFV architecture. 

Alhebaishi et al. [50] use a deployment model to track vulnerabilities across levels; however, they do not consider details of those levels. The authors in Shameli-Sendi [51] consider the efficient configuration of service function chaining to build secure customer networks using network security patterns. 

The surveys in [7,46,47,48] are very recent (2018–2021) and very comprehensive; they can be used to put our work in perspective. [7] mentions some of our work and indicates that our misuse patterns are the first ones for NFV. Also based on these surveys and reading some papers we could see that none of the other works present misuse patterns or precise models of the architecture; they just use the block diagrams of the ETSI publications. In fact, the ETSI publications about security [40,41] use only block diagrams, while we use UML which is much more precise. We have studied the references above and others to obtain the extra details. 

## 6. Conclusions and Future Work

NFV is a relatively new technology, still under development. Until now, most of the industry work has concentrated on standards, product implementation, and studies of performance and network deployment. When more NFV networks are in use we will see their attacks increasing. There has been considerable work on security but not much on modeling aspects, necessary for improving their security and use. As indicated in the section on related work our models are the only precise models in the literature, and we have published the only misuse patterns and SRA for NFV until now.

We have presented here a security reference architecture that shows where the important threats to the architecture can be expected and indicates appropriate defenses. This SRA, although partial, can be used by product developers and researchers to implement defenses and study how to improve the security of NFV networks. We have also shown the value of an SRA to add security to a complex system. While it is possible to add security defenses using lists of defenses like those in the ISO standards, the SRA makes explicit where in the architecture the defenses should be applied. Our SRA is based on previously published patterns. 

Future work can focus on extending the SRA by writing new misuse patterns that can discover the need for other defenses in specific places of the architecture. A more complete catalog of misuses would increase the practical value of this SRA. NFV is used in the new 5G standards and our models can be of value for their development. 

## Figures and Tables

**Figure 1 sensors-22-03750-f001:**
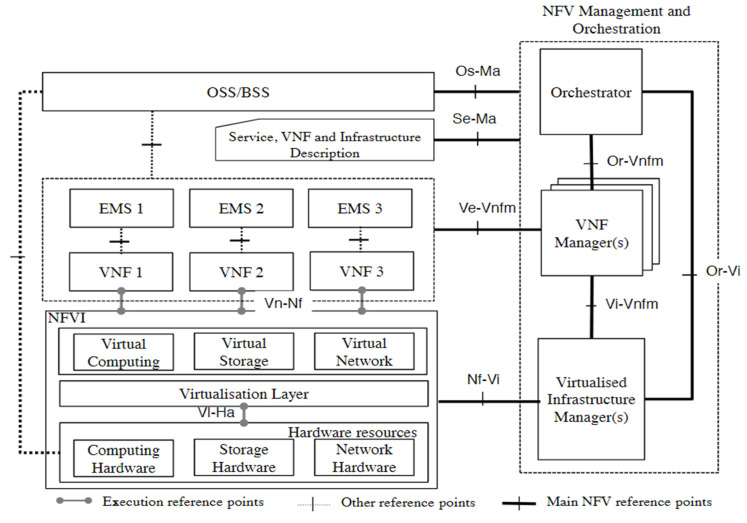
High-level NFV framework [3].

**Figure 2 sensors-22-03750-f002:**
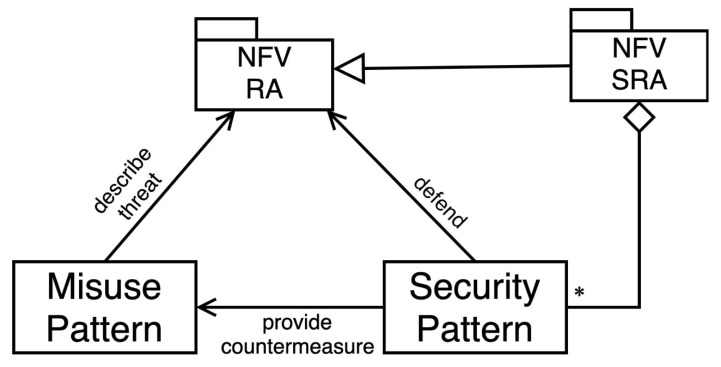
Pattern diagram for the NFV SRA.

**Figure 3 sensors-22-03750-f003:**
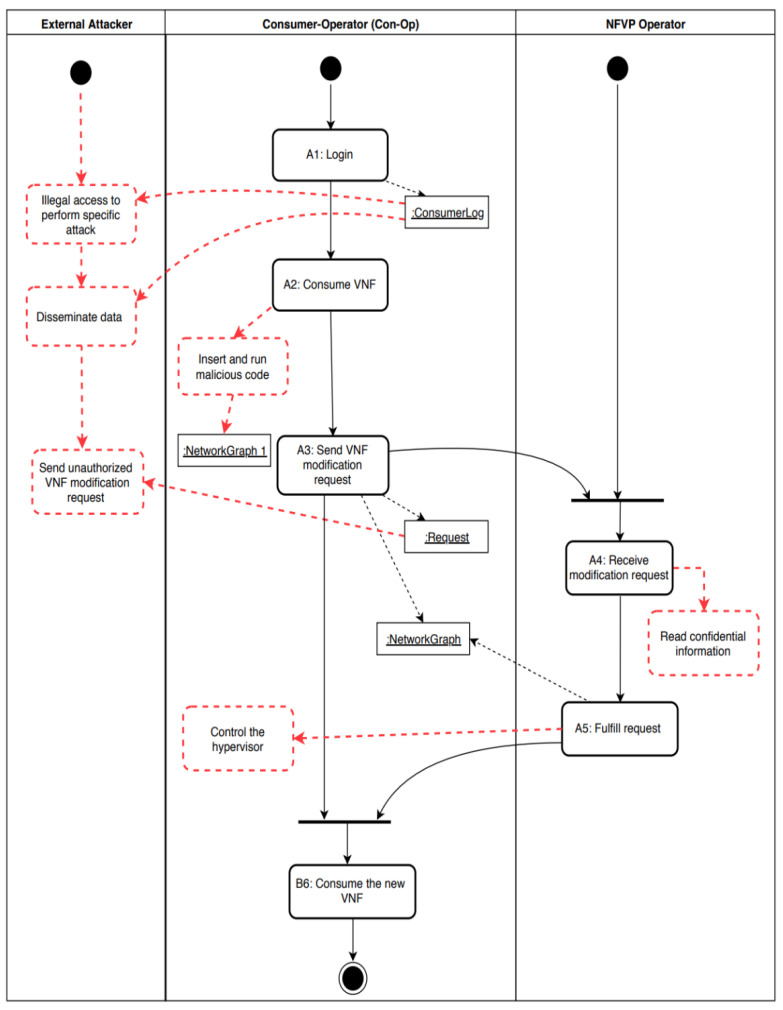
Activity diagram for UC13 “Request Modify VNF” and UC20 “Consume a Network”.

**Figure 4 sensors-22-03750-f004:**
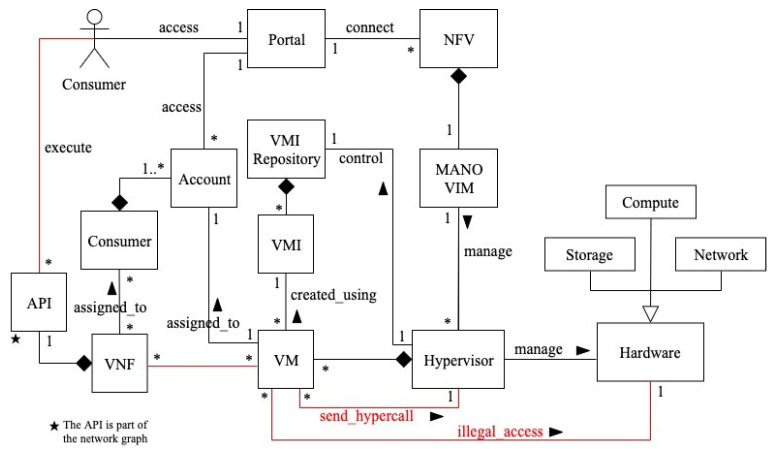
Class diagram for privilege-escalation-based misuses in NFV [11].

**Figure 5 sensors-22-03750-f005:**
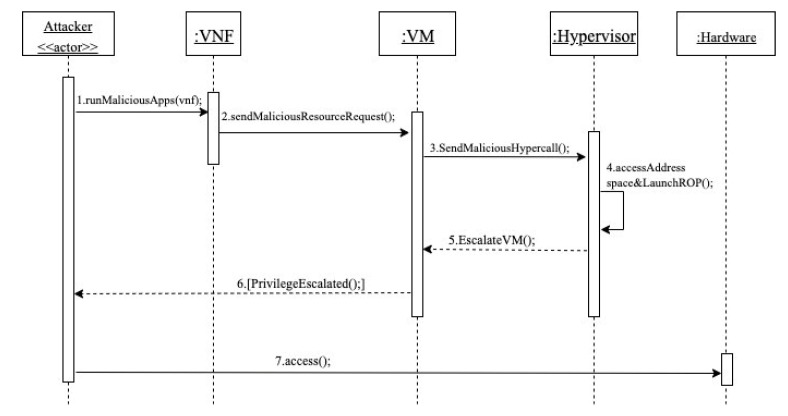
Sequence diagram for MC1 unauthorized access to hardware resources based on privilege escalation of an attacker’s VM [11].

**Figure 6 sensors-22-03750-f006:**
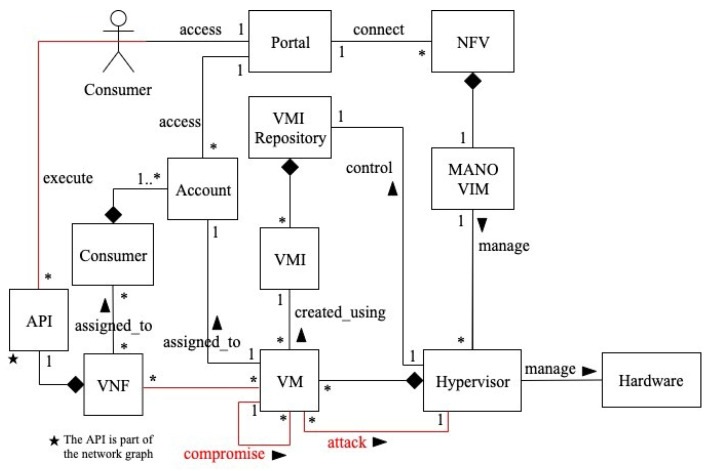
Class diagram for compromising VM via VM escape in NFV [12].

**Figure 7 sensors-22-03750-f007:**
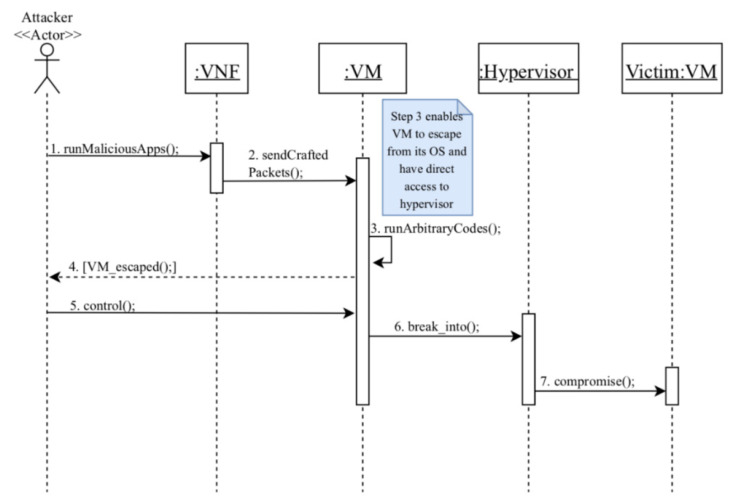
Sequence diagram for MC2: compromise a victim’s VM via VM escape [12].

**Figure 8 sensors-22-03750-f008:**
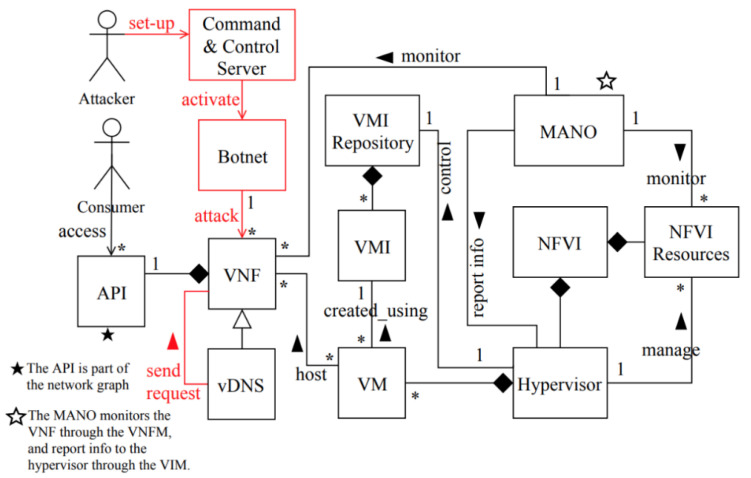
Class diagram for DDoS attack in NFV [13].

**Figure 9 sensors-22-03750-f009:**
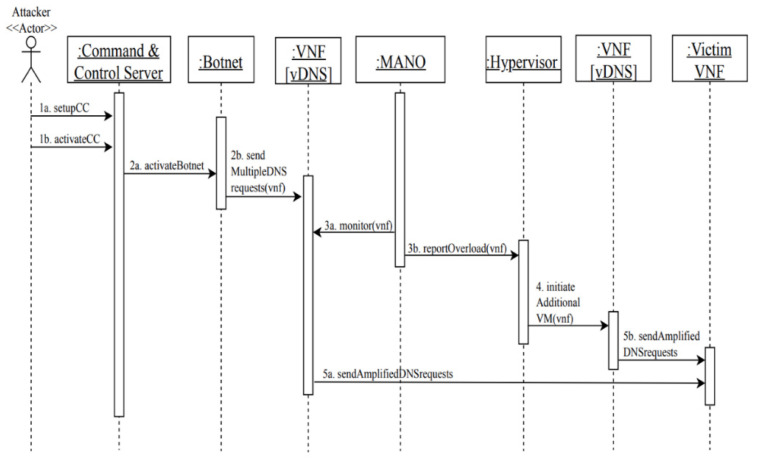
Sequence diagram for MC3: distributed denial-of-service attack in NFV using DNS amplification attack.

**Figure 10 sensors-22-03750-f010:**
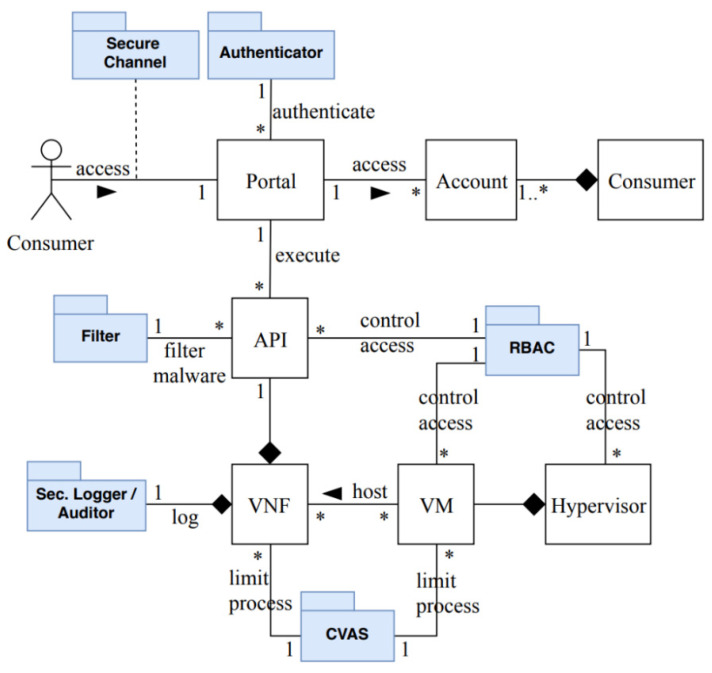
A partial security reference architecture for NFV.

**Figure 11 sensors-22-03750-f011:**
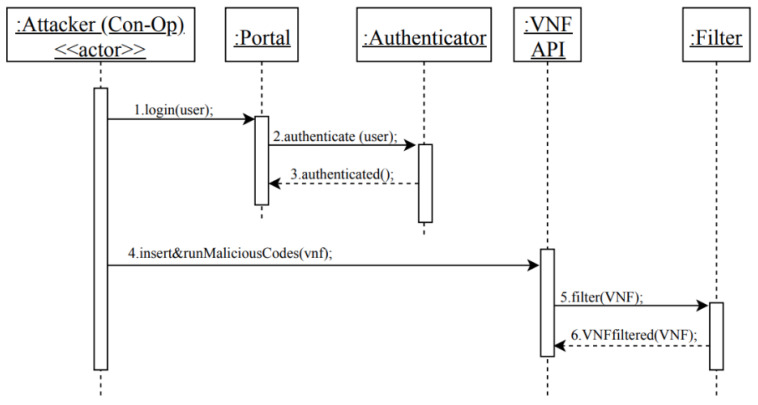
Sequence diagram for securely consume a network in NFV.

**Table 1 sensors-22-03750-t001:** Misuse activities analysis.

Actor	Action	Threat Number	Sec. Attr. CO/IN/AV/AC	Source AIn/UIn/Out	Description
Con-Op	A1. Login	T1	CO	Out	An external attacker illegally accesses the system to perform specific attacks, such as modify the network
		T2	CO	Out	An external attacker disseminates Con-Op’s data.
	A2. Consume VNF	T3	IN	AIn	The Con-Op inserts and runs malicious codes within the VNF.
	A3. Send VNF modification request	T4	IN	Out	An external attacker modifies the Con-Op requests while in transit.
		T5	AV	AIn/Out	The Con-Op overwhelms the system with requests to make the service unavailable (DoS).
NFVP Operator	A4. Receive modification request	T6	CO	AIn/UIn	The NFVP Operator collects confidential information.
	A5. Fulfill request	T7	CO/IN	AIn	The Con-Op compromises the hypervisor in unauthorized manner, which results to accomplish the attacker’s malicious goals, such as controlling the hypervisor, modify VM privilege, or VM escape.

**Table 2 sensors-22-03750-t002:** Threats list and its mitigation defenses.

Threat Number	Description	Defense Mechanism
T1	An external attacker illegally accesses the system to perform specific attacks, such as modify the network.	Authenticator
T2	An external attacker disseminates Con-Op’s data.	Security logger/auditor
T3	The Con-Op inserts and runs malicious codes within the VNF.	Filter module
T4	An external attacker modifies the Con-Op requests while in transit.	Secure channel
T5	The Con-Op overwhelms the system with requests to make the service unavailable (DoS).	CVAS, traffic filtering and detection mechanisms
T6	The NFVP operator collects confidential information.	Security logger/auditor
T7	The Con-Op compromises the hypervisor in unauthorized manner, which accomplishes the attacker’s malicious goals, such as controlling the hypervisor, modifying VM privilege, or VM escape.	Patching the hypervisor

**Table 3 sensors-22-03750-t003:** Validation of the Security Reference Architecture of NFV.

	Components	VNF	VM	Hypervisor	API	VIM	Authenticator	Filter	Sec. Log/Auditor	RBAC	CVAS	Sec. Channel
Frameworks												
The NFV SRA												
NFV SRA		✓	✓	✓	✓	✓	✓	✓	✓	✓	✓	✓
Industry NFV Frameworks Used to Compare with our SRA												
ETSI Security Framework		✓	✓	✓	✓	✓	✓	✓	✓	✓	✓	✓
CSA Security Framework		✓	✓	✓	✓	✓	✓	✓	✓	✓	✓	✓
Red Hat Platform		✓	✓	✓	✓	✓	✓	✓	✓	✓	✓	✓

## Data Availability

Not applicable. The study does not report any data.

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
