# Peer review of "Towards a Security Reference Architecture for NFV"

_sensors, 2022, doi:10.3390/s22103750_

Round 1

Reviewer 1 Report

  1. In Section 5 "related work", there is a lack of sufficient analysis of existing work on NFV safety or risk, so it is difficult to evaluate the innovation of this article.

  2. In Section III, more descriptions of NFV Security Reference Architecture should be added, and a better explanation of how SRA can deal with the above security threat examples should be given

  3. Most of the security threat use cases described in Chapter 3 are at the hypervisor or virtualization level, and the above risks also exist in cloud computing and other environments. The author should explain the particularity of the above security risks in the NFV environment
  4. More experiments are needed in Chapter 4  to verify the effectiveness of SRA
  5. Relevant work in recent two years should be added to the references

Author Response

First, we thank the referees for their comments that were very useful to improve the paper.

In Section 5 "related work", there is a lack of sufficient analysis of existing work on NFV safety or risk, so it is difficult to evaluate the innovation of this article.

--- The section on Related Work has been completely rewritten. It now shows how we verified that our work was original and puts our results in perspective. Safety is about  avoiding catastrophic situations, it is a different quality factor than security. Risk is the evaluation of the probability and impact of threats; a stage of risk analysis can be done after enumerating threats but the way to do that is not special for NFV, we added a sentence at the beginning of Section 3 to explain this point.

In Section III, more descriptions of NFV Security Reference Architecture should be added, and a better explanation of how SRA can deal with the above security threat examples should be given.

---We added more details in the beginning of Section 3 and in subsection 3.1. With the new explanation, one example should be enough to show the idea.

Most of the security threat use cases described in Chapter 3 are at the hypervisor or virtualization level, and the above risks also exist in cloud computing and other environments. The author should explain the particularity of the above security risks in the NFV environment

---Threats in clouds and edge devices have been studied in other works; including one of our own: Keiko Hashizume, David G. Rosado, Eduardo Fernández-Medina, Eduardo B. Fernandez, “An Analysis of Security issues for Cloud Computing”, Journal of Internet Services and Applications 2013, 4:5 (27 February 2013), https://doi.org/10.1186/1869-0238-4-5

We circumscribed our work only to the security of the NFV components. We added a paragraph in the Introduction to explain this point.

More experiments are needed in Chapter 4 to verify the effectiveness of SRA

--- As we indicate in Section 3.5.1, an SRA is a prototype for concrete architectures, it is not evaluated by experiments but by its completeness (it has all the relevant features), its understandability (which allows convenient application), and by comparison with industry architectures.

Relevant work in recent two years should be added to the references

--- The section on Related Work includes several new references; in particular, we added references to surveys that in turn include many more references.

Reviewer 2 Report

  • The relevance of the SRA should be demonstrated through its application to real virtualised networks, such as 5G core architectures, threats and misuse patterns.
  • Raws 198-199: duplication. "security pattern..."
  • Raws 369-370: duplication. Remove ittem 8. from the description and leave it in the post-condition.
  • Fig.10: substitute Fig. 10 with and extended version of Figure 1 with security measures.
  • Raw 689: "...ous SRA model is fairly accured..."
  • Raw 701: geneal

Author Response

First, we thank the referees for their comments that were very useful to improve the paper.

The relevance of the SRA should be demonstrated through its application to real virtualised networks, such as 5G core architectures, threats and misuse patterns.

---5G networks include NFV, but as shown in [Madi], a 5G system includes several layers with different technologies and is much more complex than just NFV networks; they have interlayer threats for example. Also, NFV does not have to be used only in 5G networks. We added a sentence about this point in the Introduction.

--Figure 1 is block diagram, and figure 10 is more precise.

Reviewer 3 Report

This paper first introduces the advantages of Network Function Virtualization (NFV) technology, but it increases the complexity of the system and greatly increases the threat of attack. Therefore, this article begins with a threat analysis of some of the NFV use cases. Use cases are scenarios in which threats to the architecture can be enumerated. Threats are described as misuse cases to describe how attackers operate, and finally, countermeasures can be found in the form of security patterns, and this paper constructs a Security Reference Architecture (SRA).

This paper analyzes the security requirements of NFV and deduces the SRA from the analysis of the threat, which has a certain reference value for the subsequent research on the security architecture of NFV, but the paper needs some improvements before publication. My detailed review is as follows:

1. This paper has little analysis of the research status of NFV security architecture and lacks a comprehensive, accurate, in-depth, and concise summary of domestic and foreign research results related to this research direction.

2. This paper proposes an idealized model with many theoretical statements but lacks comprehensive proof of the practicability, reliability, and security of this secure reference architecture. For example, after analyzing use case activities and finding different misuses, appropriate mitigation mechanisms are given, but there is no relevant proof of whether the countermeasures are effective.

3. Some grammar and spelling errors need to be checked carefully.

Author Response

First, we thank the referees for their comments that were very useful to improve the paper.

This paper first introduces the advantages of Network Function Virtualization (NFV) technology, but it increases the complexity of the system and greatly increases the threat of attack. Therefore, this article begins with a threat analysis of some of the NFV use cases. Use cases are scenarios in which threats to the architecture can be enumerated. Threats are described as misuse cases to describe how attackers operate, and finally, countermeasures can be found in the form of security patterns, and this paper constructs a Security Reference Architecture (SRA).

This paper analyzes the security requirements of NFV and deduces the SRA from the analysis of the threat, which has a certain reference value for the subsequent research on the security architecture of NFV, but the paper needs some improvements before publication. My detailed review is as follows:

  1. This paper has little analysis of the research status of NFV security architecture and lacks a comprehensive, accurate, in-depth, and concise summary of domestic and foreign research results related to this research direction.

---The section on Related Work has been completely rewritten. It now shows how we verified that our work was original and puts its contribution in perspective.

  1. This paper proposes an idealized model with many theoretical statements but lacks comprehensive proof of the practicability, reliability, and security of this secure reference architecture. For example, after analyzing use case activities and finding different misuses, appropriate mitigation mechanisms are given, but there is no relevant proof of whether the countermeasures are effective.

---Security patterns are defined with the purpose to control specific attacks. All the patterns we use have been validated by their presentation in pattern conferences and in practice. If a threat is identified in a reference architecture and that threat can be handled by a pattern we consider that if the pattern is instantiated in the architecture it will be able to stop the corresponding attack. The analysis of all the use cases would give us confidence that all the important threats have been identified. Misuse cases indicate where to apply a security pattern to stop the attack. Fig 11 shows how the insertion of security patterns can stop an attack. The use of patterns is common in software development and there are catalogs that make their use convenient [5]. Reliability is outside our scope, its methods and solutions are different from security

  1. Some grammar and spelling errors need to be checked carefully.

---The paper has been carefully revised to correct these errors.

Reviewer 4 Report

The authors present a Security Reference Architecture based on Network Function Virtualization that indicates appropriate defenses, as an answer to various possible threats. As the authors claim, the proposed architecture is partial, however, it can give a certain merit to product developers and researchers to improve the security of NFV networks. The paper is well written, however, it would be beneficial for the readers to include also the Software Defined Networks concept, as it is interlinked with NFV. Also, the authors have to "clean" the paper from the formatting remains.

Author Response

First, we thank the referees for their comments that were very useful to improve the paper.

The authors present a Security Reference Architecture based on Network Function Virtualization that indicates appropriate defenses, as an answer to various possible threats. As the authors claim, the proposed architecture is partial, however, it can give a certain merit to product developers and researchers to improve the security of NFV networks. The paper is well written, however, it would be beneficial for the readers to include also the Software Defined Networks concept, as it is interlinked with NFV. Also, the authors have to "clean" the paper from the formatting remains.

---SDN is an implementation technology which is often used in NFV, but it is independent of it. Security in SDN has been studied in several works but we wanted here to separate clearly the threats due to NFV. We added a paragraph in the introduction to indicate this point.

Round 2

Reviewer 1 Report

.

Reviewer 3 Report

This paper is in the position for acceptance now.